# Engineering Organ-on-a-Chip to Accelerate Translational Research

**DOI:** 10.3390/mi13081200

**Published:** 2022-07-28

**Authors:** Jihoon Ko, Dohyun Park, Somin Lee, Burcu Gumuscu, Noo Li Jeon

**Affiliations:** 1Department of Mechanical Engineering, Seoul National University, Seoul 08826, Korea; jihoonxx@snu.ac.kr; 2Bio-MAX Institute, Seoul National University, Seoul 08826, Korea; parkkhdh@gmail.com; 3Interdisciplinary Program in Bioengineering, Seoul National University, Seoul 08826, Korea; sml1003134@gmail.com; 4Biosensors and Devices Laboratory, Biomedical Engineering Department, Institute for Complex Molecular Systems, Eindhoven Artificial Intelligence Systems Institute, Eindhoven University of Technology, 5600 MB Eindhoven, The Netherlands; b.gumuscu@tue.nl; 5Institute of Advanced Machines and Design, Seoul National University, Seoul 08826, Korea

**Keywords:** organ-on-a-chip, microfabrication, microphysiological system, biophysical stimuli, biochemical stimuli, in vitro cell culture

## Abstract

We guide the use of organ-on-chip technology in tissue engineering applications. Organ-on-chip technology is a form of microengineered cell culture platform that elaborates the in-vivo like organ or tissue microenvironments. The organ-on-chip platform consists of microfluidic channels, cell culture chambers, and stimulus sources that emulate the in-vivo microenvironment. These platforms are typically engraved into an oxygen-permeable transparent material. Fabrication of these materials requires the use of microfabrication strategies, including soft lithography, 3D printing, and injection molding. Here we provide an overview of what is an organ-on-chip platform, where it can be used, what it is composed of, how it can be fabricated, and how it can be operated. In connection with this topic, we also introduce an overview of the recent applications, where different organs are modeled on the microscale using this technology.

## 1. Introduction

Organ-on-a-Chip (OoC) can be described as a microfluidic device with small structures used for culturing cells. The small structures define the locations of cell growth, or create biochemical (e.g., growth factor) and biophysical (e.g., electrical, thermal, and mechanical) stimuli for generating physiologically relevant microenvironments. Therefore, OoC is an integrated system of microfluidic and bioengineering technology designed to reconstitute the tissue architecture of specific organs and recapitulate a key physiological function of a tissue. As a preclinical in-vitro model, testing the efficacy or toxicity of drugs and studying the pathophysiology of the tissue are two major applications of OoC. This novel experimental model with high human physiological relevance is built to compensate for the limitations of traditional 2D petri dish culture or animal models, narrowing the gap between preclinical and clinical results in drug developments and disease studies. The key functions of various organs, tissues, and pathologies have been modeled using OoC since the early 2010s [1]. Since then, the advancement in the field of OoC technology has accelerated and many novel models were reported as proof-of-concept studies in academics. Table 1 summarizes the advantages and disadvantages of the techniques being used to study cell microenvironments.

The use of OoC has just begun to expand toward clinics and pharmaceutical industries [2,3]. Along with the emerging interest in personalized and precision medicine as well as the advent of novel therapeutics such as cancer immunotherapy, the biomedical community started to positively consider the adoption of OoC as a promising tool for biomedical and pharmaceutical research [4]. Although technical and biological improvements are still needed to fulfill the criteria of its routine and universal use, global efforts from diverse professional fields started developing and applying this novel technology [3]. Based on this current tide in OoC community, recently published review papers focus on sharing opinions on what this society should pursue when developing the model successfully applied on preclinical study or drug discovery [5,6,7]. In this paper, we start by describing the fundamentals of the OoC, such as basic and advanced engineering techniques and biological components inside the OoC followed by the representative applications of OoC. This will guide the researchers from non-OoC community and provide better understanding on the potentials and advantages of OoC. Furthermore, the article will discuss about the current position and limitation of OoC in translational research and its mission and future perspective on successful bench-to-bedside cases.

## 2. Key Features of OoC

OoC is a microengineered device to recapitulate key functions of organs and tissues. In these platforms, the basic idea is not to generate organs themselves, but the functions of the organs are mimicked to study a particular aspect of the target organ. OoC platforms were invented for mitigating the limitations of both conventional cell culture plates and animal models (see Table 1 for an overview of the limitations). Although animal models can help us to understand various biological phenomena better, they are usually too complicated for mechanistic studies. Compared to in-vitro models, animal experiments are costly and time-consuming and it is challenging to observe phenomena happening in the deep tissue of the models. On the other hand, conventional cell culture plates allow- us to control the types and numbers of cells as well as apply controllable stimuli. Yet, integration of other relevant systems such as endocrine, neurological, and immunological considerations is still lacking in conventional cell culture plates. OoCs can provide well-defined, well-controllable, easy to observe, but still complex environments to display organ-specific functions. All OoC platforms have three fundamental characteristics: (i) the arrangement of cells in in-vivo-like layouts; (ii) the possibility to culture multiple cell types to reflect physiological relevance; and (iii) the presence of biochemical and biophysical stimulations to mimic the functions of tissues. To realize these characteristics, several technical and practical elements related to microfabrication and cell biology are applied, including the design, materials, the fabrication method of the platform, cell sources, the type of scaffold, and the type of the stimulus (Figure 1). In this paper, OoCs will be introduced in the context of these three characteristics and technical factors for producing these platforms.

## 3. Construction of an OoC Platform

Multiple cells interact with each other within micrometer-scaled environments in OoC systems. To build such systems, several microfabrication strategies can be used. Since the introduction of the first OoC device in the 2010s, the vast majority of OoC systems were fabricated using soft lithography [8,9]. Lithography is a combination of two words: “lithos” which means “stone”, and “graphein” which means “to write” in Latin. Soft lithography is a process where a microchip layout is engraved in a soft material. In most cases, this soft material is polydimethylsiloxane (PDMS). Having several key properties including biocompatibility, optical transparency, gas permeability, and high-definition patternability make PDMS ideal for OoC and biological applications. Furthermore, the elasticity of PDMS enables OoC devices to utilize biophysical and mechanical stimuli to recapitulate the function of an organ. Fabrication of a PDMS-based OoC device starts with the design of the desired microchip layout in the software, such as AutoCAD, and transferring this layout to a photomask [10,11]. A photomask is an opaque surface printed on glass or film with transparent spots, or patterns, to allow light to pass through in a defined pattern. The photomask is used to fabricate a mold master. The mold master is usually made with photoresist upon a silicon substrate via a photomask and ultraviolet light exposure. A photoresist is a light-sensitive polymer, changing its molecular structure upon exposure to ultraviolet light and turning it into a soluble or insoluble material. A common photoresist type used in microfabrication is called SU-8. The solubility effect depends on the tone of the photoresist and in this way, the photomask with features can be replicated on a mold master with a positive or negative polarity. PDMS is a liquid polymer in the uncured form, upon mixing it with an activator solution and subsequently applying heat, it solidifies thanks to cross-linking reactions. Therefore, PDMS in the liquid form is poured on top of the mold master, and exposed to heat to structure the microchip layout into the PDMS in the solid or cross-linked form. The PDMS-based OoC device with a microchip layout is then punched to open all through holes that will be used for liquid injection ports (such as to inject cell culture media into the device). The final step is to bond the casted PDMS OoC device with a glass or other PDMS part via plasma bonding. Oxygen plasma exposed to the PDMS surface renders silanol groups (–OH) on the surface and immediate meeting of the surface with other oxidized PDMS or glass forms an irreversible Si–O–Si bond at the interface. This covalent bond prevents water from permeating into the glass-PDMS interface. Bonding of an engraved PDMS OoC device with a flat surface forms a microchannel where a hydrogel-cell mixture, cell suspension, or culture medium can be inserted. Such a microchannel is a basic design factor of the OoC devices. Figure 2a shows an example of the fabrication process of a PDMS-based device; the lung-on-a-chip device [12].

Although PDMS is an ideal material for OoC operations, it also has disadvantages [13,14]. For example, PDMS can absorb small and hydrophobic molecules. This hinders the use of PDMS-based OoCs from drug studies as the actual dose of drug introduced to cells will not be clear due to the absorbance effect. Apart from that PDMS is a soft and flexible material, which can be undesired for some of the applications requiring hard materials.

Another way of fabricating OoCs is based on injection molding (Figure 2b) [15,16]. This technique is mostly preferred in commercialized OoC systems because it allows for mass production. Injection-molded materials are mostly plastics, and therefore they are not flexible and do not absorb or permeate other molecules, but they are still optically transparent. Injection molding is a microfabrication process adapted from the industry. The molten material is injected into a previously machined mold, and the material gets solidified upon cooling down. The type of materials used in this process involve thermoplastics, elastomers, and polymers. Microstructured plastic parts are integrated into complete OoC devices by assembling with strews, solvent bonding, or thermal bonding [15]. Other bonding methods such as silicone adhesive bonding and plasma bonding with PDMS can also be used according to the materials selected to fabricate the main body. Some injection-molded OoC devices utilize the dimensions of conventional well-plates to be compatible with conventional liquid handlers or readout systems [17,18,19,20,21,22].

Another preferred fabrication method is 3D printing [23,24]. This technique offers the advantage of precise control over geometry on the microchannel geometry. Fabricating circular microchannels is challenging via soft lithography because the photoresist can be spread on a silicon substrate and crosslinked with a fixed height only. 3D printing techniques can control the spatial distribution via layer-by-layer assembly of the material [25]. It is also possible to print the cells in 3D on a substrate along with an extracellular matrix such as a hydrogel [26]. Printing ink can be made of natural (hydrogel, collagen, fibrin, alginate, etc.) or synthetic (polycaprolactone, silicone, gel-like Pluronic 127, etc.) material [27]. These inks can be 3D printed using several methods including (i) micro-extrusion printing, where the ink is directly deposited onto a substrate by using a micro-extrusion head, and (ii) inkjet printing, where the ink is distributed from an electrically heated or piezoelectric actuator nozzle and deposited onto a substrate in droplet form, and (iii) laser-assisted printing, where a laser beam writes on an ink coated on a substrate. The final resolution is the best when laser-assisted printing is used, while inkjet printing is a preferred technique to print the cells together with their supporting matrix. Complex architectures in human organs can be replicated more precisely, allowing for better recapitulation of the key tissue and organ-level functions. The 3D printing technique is also applicable for commercialized products and large-scale production of OoCs and, therefore, it also finds a place in the industry. Figure 2c demonstrates 3D printing methods to fabricate microfluidic devices with complex-shaped microchannels.

OoC devices can be fabricated using various techniques as summarized in this section. The fabrication method and material are selected according to the purpose of the study in the design step. PDMS is a suitable material to fabricate a few micrometer-scaled microchannels and 3D printing can be preferred to fabricate a microchannel network to mimic a complex 3D vascular structure.

**Figure 2 micromachines-13-01200-f002:**
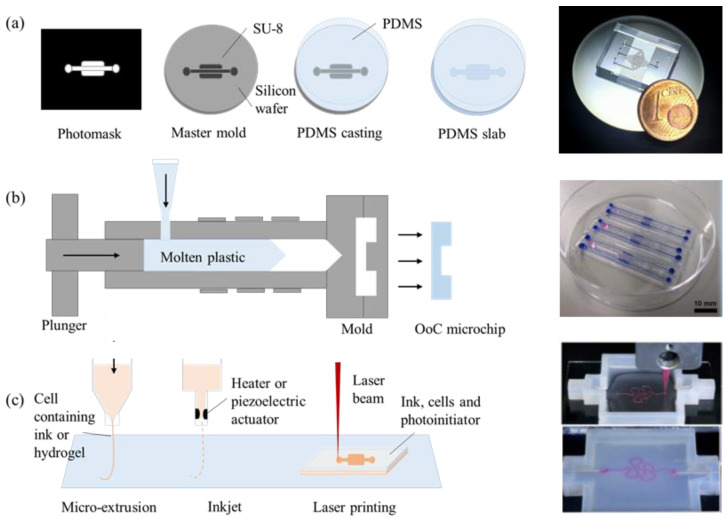
Fabrication methods of organ-on-a-chip models. (**a**) Schematic process of fabricating PDMS-based microfluidic devices. Microchip figure (**right**) reproduced from Ferraz et al., 2018 [11]. (**b**) Schematics and images of an injection molded microfluidic device. Microchip figure (**right**) reproduced from Virumbrales-Munoz et al., 2019 [16]. (**c**) Schematics and images of microchannels fabricated by a 3D printer. Complex-shaped channels can be fabricated with a 3D printer. Microchip figure (**right**) reproduced from Homan et al., 2016 [24].

## 4. Cell Microenvironments Mimicking In-Vivo

The first consideration in designing OoC devices is to arrange cells to exploit their key roles in organ function. Microchannels or pores are utilized to confine cells in a target area. Cells can be confined into droplets in droplet-based microfluidic devices or can be confined onto a surface made of hard (e.g., plastic) or soft (e.g., hydrogels) materials while this system is fed by parallel or coaxial flows. In the former case, the microfluidic device has at least one droplet-generator unit (commonly in the form of T, X, Ψ, or Y-shaped junctions) and a droplet splitting and merging unit for droplet handling (Figure 3a) [28,29,30]. Droplets are formed from a dispersed phase fluid into compartmentalized droplets surrounded by a continuous phase fluid. Droplets can be produced in various sizes and monodisperse (uniform size) forms and they are useful templates for drug delivery, nutrient delivery, and living cell encapsulation [31]. In the latter case, microchannels are engraved into a solid block of microchips, which may contain microgrooves with different heights, membrane-like barriers, micromechanical valves, and various micro-scale components [1,32,33]. Cells are seeded and grown in the vicinity of these structures depending on the OoC application. Parallel-flow-based microchips are useful models for external stimuli generation and spatial control of the cell culture layout. For example, neurons can be grown in microchips. Microchannels with 10 μm of width and 5 μm of height confined can prevent somas from intruding into the channels and only allow axons to grow along the channels, resulting in the perfect isolation of axons [33]. Similarly, a liver sinusoid model was built using a microfluidic endothelial-like barrier consisting of a parallel array of channels with a width of 2 μm and height of 1 μm [32]. The barrier was used for forming high resistance into the cell culture area and it also plays a key role in concentrating hepatocytes within the culture chamber, where the concentrated hepatocytes showed higher viability compared to the hepatocytes cultured in low density.

Capillary action is often used as a strategy to build the cellular microenvironment in OoC devices. Capillary action is governed by the interplay between the geometry or chemistry of a surface and the surface tension of a liquid. In microchips, capillary action ensures the spatial control of the liquid, i.e., liquids can be selectively patterned in desired spots. Micro bumps or micro pillaris are examples of elements to control the flow via surface geometry (Figure 3b,c) [34,35,36,37]. Microbumps and micropillars in the microchannels form narrow gaps that serve as gates, allowing or blocking the flow [34]. To allow for the flow into an adjacent microchannel, high bursting pressure should be applied. Using microscale gates, a hydrogel solution containing cells can fill a microchannel and another solution can fill the adjacent microchannel after cross-linking of the previously loaded hydrogel. As the hydrogels are physically connected, the cells in both hydrogels can chemically communicate and migrate into the other zones. Controlling the flow via geometrical change strategy was used in many OoC models for cell arrangement in several applications. Examples include vascular networks, angiogenesis models, blood-brain barrier models, and tumor extravasation models [38,39,40,41]. Along with geometrical changes, surface chemistry changes can be used for arranging cells in flow-based microchips (Figure 3d). Open microfluidic devices which have additional air-liquid interfaces other than loading ports use hydrophobically modified surfaces by air plasma to induce the wicking of liquids through a narrow gap [42,43]. Hydrogel solutions introduced to a microchip tend to fill areas with high wettability or hydrophilic areas and the difference in wettability enables the allocation of cells with in-vivo-like layouts within the devices. Use of different materials within a single device, air plasma, a hydrophobic or water-repellent coating is used for controlling wettability in microfluidic devices.

**Figure 3 micromachines-13-01200-f003:**
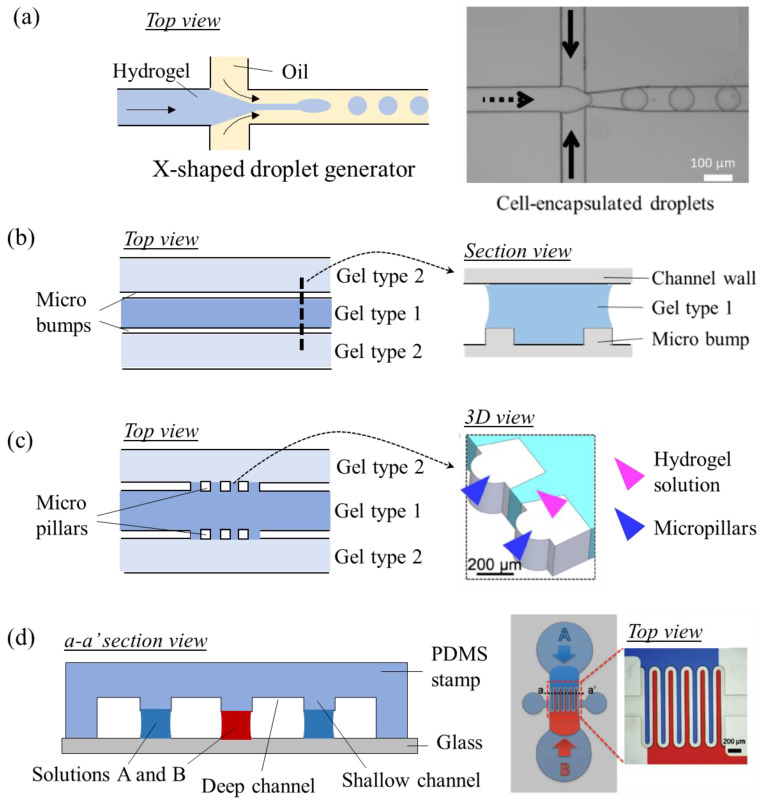
Methods used for positioning cells with in-vivo-like layouts. (**a**) Schematics of a droplet-based microfluidic device for culturing single-cells in droplets. The right panel figure is reproduced from Chan et al., 2013 [30]. (**b**) Schematics of a microchannel with micro bumps, allowing for the injection of different hydrogel types without physically dividing the channel. (**c**) Schematics of micropillars working as capillary valves. The right panel figure is reproduced from Hyung et al., 2021 [37]. (**d**) Schematics of capillarity-mediated liquid patterning. Shallow channels with hydrophilic regions attract the solution while hydrophobic regions repel the solution with water content. Shallow channels guide cells to be patterned in designed shapes. Figure reproduced from Lee et al. (2010) [43].

## 5. Selection of Cell Resource

The next critical element of an OoC device is the choice of cell resources. The same cell study performed in commercially available OoC devices may yield genetically and phenotypically different results if the cells from different resources are used [44]. Hence, cell resources should be selected according to the physiological relevance, sustainability, and purpose of the study. The most widely used cell resources are commercialized cell lines, primary cells, and induced pluripotent stem cells (iPSCs) from trusted supplier companies and cell banks.

Commercial cell lines often represent immortal cell lines that are easy to culture using established and reliable culture protocols. These cell lines grow in predictive ways, resulting in reproducible OoC models e.g., for optimization purposes. As commercial cell lines are popular in biological studies, there is vast information available in the literature about the morphology, gene profile, and molecular pathways belonging to the cell type of interest. Users, accordingly, take advantage to use such information to design experiments or compare the results observed in an OoC model. Some commercially available cells can be excellent sources of rare cell types that are difficult to differentiate from iPSCs. However, mutations originating from immortalization or multiple passages can decrease the physiological relevance of the OoC models when compared with the primary cells obtained from donors.

Primary cells directly obtained from donors are an excellent source when developing donor-specific OoC models. The National Center for Advancing Translational Sciences (NCATS; https://ncats.nih.gov/tissuechip, accessed on 1 August 2020) in the United States initiated a grant program to develop Clinical Trials on a Chip in 2020. The majority of granted proposals aimed at developing patient-specific OoC models, also called personalized medicine, using patient-derived cells. Due to the heterogeneity of patients, especially cancer patients, direct use of primary cells is an ideal approach to reconstitute diseased conditions within OoC devices, and it is believed that these cells can repeatedly generate responses against therapeutics. Even though primary cells have advantages in an identical gene profile to the donor, it is challenging to reuse or store the primary cells in the longer term (in the order of several weeks). When working with these cells in ex-vivo cultures, great care in isolation protocol, the composition of the cell culture medium, and the design of the microcellular environment are required to retain their original phenotypes.

Adult stem cells or iPSCs are increasingly used in OoC platforms because this cell type is infinitely sustainable and can be differentiated into various tissues, including diseased tissues via genetic engineering. For example, organoids representing different tissues derived from a single donor can be co-cultured in a multi-organ chip platform to show the systematic connection of organs. iPSC cells can be used to create fully developed tissues from scratch, although such studies require extensive time and resources. So far, the use of iPSCs has remained limited in developing OoC models to test therapeutics in a patient-specific manner because the protocols for differentiation and maturation are not standardized and have low reproducibility levels.

While commercialized cell lines are easy to access, they are not ideal for OoC studies requiring a high correlation with real tissues. Primary cells and iPSCs are becoming increasingly popular in OoC studies due to having a patient-specific, original gene profile and phenotype. On the other hand, these cells are difficult to access due to the non-frequent supply through biopsies or surgeries. Another drawback is that the differentiation of iPSCs is a time-consuming and laborious process.

## 6. Application of Stimuli

The ability to apply mechanical stimuli to cells is one of the most important features of OoC platforms. OoC platforms can generate precisely controlled mechanical stimuli that were challenging to be applied to conventional models. The capability of OoC platforms to apply a mechanical stimulus to the cells facilitated observing the response in a 3D environment and in real-time. This enabled the regulation of the growth of cells in OoC platforms and to recapitulate organ function in a more in-vivo-like fashion. This chapter describes how OoC platforms generate mechanical stimulus by exemplifying stretching, compression, flows, and shear stress as displayed in Figure 4.

Many OoC platforms utilized the elasticity of PDMS for generating mechanical stimuli of stretching and compression. A representative example of stretching is the lung-on-a-chip device [1]. The device consists of a middle channel vertically separated into two sections mediated by a porous membrane and two side channels on both sides of the middle channel. Negative pressure applied to the side channels deforms the thin walls of the middle channel, resulting in the stretch of the membrane in the middle channel. The inventors applied 10% of cyclic strain to the membrane where lung epithelial cells and microvascular endothelial cells are attached on each side respectively. The presence of cyclic strain significantly enhanced transmembrane uptake of nanoparticles into the endothelial layer as large as mouse model uptake. The article showed the importance of mechanical stress in modeling organs and demonstrated the usefulness of OoC devices to generate mechanical stress.

Also using the elasticity of PDMS, some OoC devices demonstrated responses of tissues against compression. Magdesian et al. demonstrated injury neuronal network by pressing the top of OoC device where neurons are cultured with a beaded AFM tip and observed reconnection of the network [45]. Ahn et al. demonstrated injury of microvessels induced by compression using an OoC platform [46]. An air channel is fabricated above a microvessel channel and the thin layer of PDMS between the two channels deforms to press the vessel zone when positive pressure is applied to the air channel. The compression is expected to be used for mimicking pressure-induced diseases such as glaucoma and asthma.

Various types of fluid flows exist in our body such as interstitial flow, intravascular flow, and transendothelial flow and they play essential roles in the differentiation, proliferation, migration, and gene expressions of cells [47]. OoC platforms are efficient tools for studying the effect of flows on cells. For example, Kim et al. reported interstitial flow regulates the angiogenic response and phenotype of endothelial cells. They found angiogenic sprouting was enhanced toward the reverse direction of the interstitial flow and the sprouts display abundant actin-rich filopodia at their distal edges [48]. Recently, Hajal et al. applied both luminal flow and trans-endothelial flow to the vascular network and demonstrated the increased potential of tumor extravasation [49]. These studies utilizing OoC platforms broadened our knowledge of cell biology under flow conditions.

There are many ways to generate flows within OoC platforms. Syringe pumps, pneumatic pumps, and peristaltic pumps are mainly used to generate flows by delivering culture medium. A kidney-on-a-chip platform utilized continuous medium flow over an epithelial monolayer generated by a syringe pump, resulting in 0.2 dyne/cm of shear stress applied to the cells [50]. The shear stress enhanced cell polarization and primary cilia formation compared to a static condition. In case precise control of flow is not required, OoC models often utilize gravity-driven flows. Pressure difference induced by the difference of heights of medium between medium reservoirs generates a flow to meet the balance of the heights [51]. Sometimes pipette tips or external accessories are inserted into the medium reservoir to extend the volume of the medium. This gravity-driven flow cannot be constant as the pressure difference is continuously reduced as the medium flows. Gravity-driven flow is also used to generate a pulsatile flow by placing an OoC platform on a rocker that flips tilting periodically [52].

Other than mechanical stimulation, microfluidic chips facilitate generating biochemical stimulation such as oxygen, [53,54]. nutrient, [55]. and growth factor gradient [56]. Brennan et al. comprehensively summarized microfluidic chips that adopted oxygen control techniques and oxygen sensors. They reviewed oxygen control methods including diffusion from a source fluid, separate gas perfusion, hydration layer, cellular consumption. Gradient formation of soluble factors is one of the significant advantages of microfluidic devices. Laminar flow at merging T or Y-shaped microchannels forms a diffusive profile. Using this phenomenon, various microfluidic chips for gradient formation were developed and they were applied for studying response of cells against biochemical cues such as cancer metastasis, immune response, axon guidance, and angiogenesis.

As mentioned so far, OoC platforms can apply mechanical/chemical stimulation to the tissues cultured in the platforms. External equipment can precisely control the stimuli and sometimes the stimuli can be applied to some specific part of the tissues. The controllable and selective stimuli are making OoC platforms attractive and powerful compared to other models.

Monitoring tools for drug toxicity in the cellular microenvironment.

Spatially and temporally resolved information about cell physiology and microenvironment as well as pharmacodynamic drug responses are monitored in the form of imaging, electrical signal measurement, and molecular measurements. Real-time monitoring in OoC devices is possible at the tissue level which is very difficult to observe in-vivo and difficult to reconstitute in simpler in-vitro models.

A tissue can be tracked by high-resolution imaging, which is capable of tracking single-cell level activities using microscopes. The tracking is made possible by staining the molecules of interest in such tissue constructs. Staining is performed using commercially available biomarkers which come in a range of different fluorescent colors. Mostly used biomarkers include primary and secondary antibodies, nucleus stains, cytoskeleton stains, and biomarkers.

Real-time information about the viability and metabolic activity of the tissue constructs and organoids is also observed using electrodes. The electrodes can be embedded in a microchannel thanks to microfabrication processes. Common usage areas of the electrodes include the measurement of transendothelial electrical resistance (TEER) while multi-electrode arrays (MEA) measure field potentials of cells (for example, for neural network characterization). A good example is the microfluidic blood-brain-barrier (μBBB) model that incorporates TEER electrodes on two channels separated by a porous membrane coated with endothelial cells and astrocytes on both sides, respectively [57]. Shear stress was induced by culture medium flow through the microchannel containing endothelial cells. The presence of the shear stress increased TEER levels in OoC co-culture compared to that of transwell co-culture. Transient drop and recovery of TEER were monitored in real-time in response to histamine exposure. The OoC device was used for testing barrier-enhancing or barrier-opening drugs to regulate drug delivery into the central nervous system.

The molecular analysis presents a multi-faceted way to collect information in OoC devices [58]. The supernatant of the cells or culture medium collected from the outlet is utilized for measurements in-chip and off-chip. For off-chip measurements, extraction of cells may become tricky for the devices fabricated with a permanent bonding method. In that case, an additional process of dissolving the extracellular matrix or detaching cells from the device surface is required. For profiling genes such as RNA sequencing, a sufficient number of cells is harvested from multiple OoC platforms experimented in the same condition due to the small number of cells contained in a single chip. Integrated sensors have been the workhorse for the in-chip measurements. Several types of sensors are presented to measure culture microenvironment (e.g., pH, oxygen level, nutrient content), mechanical stimulation (e.g., flow rate, compression, stretching), electrical stimulations (e.g., neural network signaling, cardiac signaling via pulse generation), chemical gradients (e.g., chemical factors, biomarkers, cell secretome ingredients).

## 7. Applications of OoCs

OoC technology in tissue engineering applications has been proved to demonstrate several advantages. OoC devices represent; physiologically-relevant systems mimicking key functions of tissues and organs; microfluidic devices that are compatible to apply biochemical and biophysical stimuli; spatiotemporally controllable 3D cell microenvironments. These powerful functions of OoC boost our knowledge in biological and pharmaceutical research. OoC devices are useful tools that can efficiently deliver reliably reproducible results for drug toxicity assessment and large-scale preclinical trials in the drug development pipeline.

## 8. Drug Development

The drug development process consists of five phases: (1) discovery and development, (2) preclinical research, (3) clinical development, (4) FDA review, and (5) FDA post-market safety monitoring. Developing a new drug takes 7 to 15 years (an average of 13.7 years). Overall, the probability of success for a drug from phase I to approval is 1 in 10,000, while the preclinical phase has the lowest success rate with 3% [59]. Due to the low success rates, the preclinical phase is called the “Death Valley” in the pharmaceutical industry. If a drug compound passes the preclinical test, the success rate can increase dramatically. A key requirement for passing the “Death Valley” is an intermediary research model that can accurately assess toxicity and predict human responses at the preclinical stage. Traditionally, toxicity and off-target effects are assessed with in-vitro 2D cell cultures or in-vivo animal experiments before clinical trials. In-vitro 2D cell cultures are seen as insufficient tools for drug testing due to the lack of complexity. Such tools typically adopt a simplified 2D model based on immortalized cell types, while heterogeneous interactions between cells cannot be recapitulated in 2D cell cultures. On the other hand, in-vivo animal models are too complex for preclinical research due to the differences in function at various organ/tissue levels, including immune system responses. Animal models may differ in tissue function when compared to humans, that is why the results from animal models may not be used for testing all drugs. OoC devices can provide sufficient complexity in cell cultures with the help of external stimuli, recapitulate the key functions of the tissues, and facilitate working with primary cells from patients and healthy individuals. In this way, OoC systems elicit appropriate responses to drug exposure. Table 2 summarizes the differences between animal models, 2D cell cultures, and OoC devices.

Cancer therapeutics requires a deeper understanding of not only cancer but the surrounding tumor microenvironment, which is not only made of cancer cells but also stromal cells, immune cells, blood vessels, and extracellular matrix [60]. Reconstructing a tumor microenvironment in OoC devices is regarded as a valid model for anticancer drug screening. After co-culture modalities of blood vessels and tumors, including peripheral stromal cells, have been established, increasing research attempts have been made to evaluate the emerging anticancer drugs, as well as strategies for chemotherapy, targeted therapy, and immunotherapy.

Tumor-on-chip technology has been developed extensively in the past decades. For example, an injection-molded 3D array was introduced for determining cell migration. The 3D array was filled with a collagen matrix, where natural killer cells migrated toward HeLa cells; therefore, spatiotemporal observation and analysis of the interaction between immune cells and cancer cells were made possible (Figure 5a) [19]. A tumor microenvironment consisting of a blood vessel and a lymph vessel can be fabricated using 3D printing. 3D-printed tubular structures serve as perfused microchannels, where complex mechanisms in molecular transport of anticancer drugs can be profiled between the vessels. A human microcirculatory system with stimulated neutrophils was also modeled in OoC to study systemic infection. This model facilitates the monitoring of dynamic interactions between intravascular tumor cells and neutrophils at high spatiotemporal resolution. The model was able to distinguish a chemokine-dependent neutrophil migration pattern that results in enhanced tumor cell extravasation (Figure 5b) [61]. In another example, organotypic endothelial cells were constructed in the form of lumen-like structures in a hydrogel environment. Using tumor endothelial cells, a patient-specific angiogenesis assay was developed to spatiotemporally evaluate the characteristics of cancer angiogenesis in each patient with kidney cancer (Figure 5c) [62].

Developing cell spheroids and organoids has also been explored in OoC devices. Table 3 provides an overview of on-chip cancer models that are used for drug treatment. A 3D vascularized ovarian cancer spheroid was exposed to Paclitaxel (Taxol^®^) and the uptake is examined through diffusivity measurements, functional flux analysis, and accumulation of fluorescently bound drugs using OoC. The presence of interstitial flow resulted in differences in responses corresponding to shrinkage and CD44 expression of vascularized tumors to Taxol [63]. A microtumor model supported by a microvascular network study explored efficacy assessments with FOLFOX (5-FU, Leucovorin, and Oxaliplatin), as a standard treatment chemotherapy drug. Tumor growth was significantly reduced in-vitro compared to the control “placebo” group [64]. Anticancer drugs were also assessed using OoC devices by focusing on the evaluation of tumors and blood vessels. In an OoC device, angiogenesis toward a glioblastoma tumor spheroid was monitored. In this system, inhibition of angiogenesis was observed when Bevacizumab (Avastin^®^), a representative anti-VEGF drug, was introduced to the system [18]. More details about microfluidic chips for anti-cancer drug screening are reviewed in the following articles [65,66]. None of these systems were used in preclinical research in drug discovery phases yet, although the emerging technology is increasingly being commercialized by industry at the moment (Figure 6).

**Table 3 micromachines-13-01200-t003:** Examples of cancer-vascular models with anticancer drug screening.

Cancer Therapy	OoC Device	Drug Treatment	Target Region	Reference
Chemotherapy	Ovarian cancer (A549, Skov3) spheroid and blood vessel coculture	Paclitaxel (5.0 μM) On day 7 of culture	Inhibited tumor cell proliferation by increasing intrinsic apoptosis	[63].
	Breast cancer (MCF-7) spheroid and blood vessel coculture	Paclitaxel (0–500 ng/mL) On day 7 of culture		[67].
	Colorectal cancer (HCT116) cell and blood vessel coculture	FOLFOX 100 μM 5-FU; 10 μM leucovorin; 5 μM oxaliplatin On day 6–8 of culture	Inhibition of DNA synthesis in cancer cells by the formation of crosslinks in DNA	[66].
Targeted therapy	Colorectal cancer (CRC-268) cell and blood vessel coculture	Bevacizumab (10 μg/mL) On day 7 of culture	Inhibiting the binding of VEGF to	[68]
	Glioblastoma (U87MG) cancer spheroid and blood vessel coculture	Bevacizumab (1.0 mg/mL) On day 3–7 of culture	Cell surface receptors	[18]
Immunotherapy	Glioblastoma (patient sample), blood vessel and tumor- associated macrophage (TAM) coculture	Nivolumab (1 μg/mL) BLZ945 (0.1 μg/mL)	PD-1 blockade (Nivolumab) CSF-1R inhibitor (BLZ945)	[69]

## 9. OoC Devices Used for Drug Toxicity Assessment

Toxicity to human tissues and unknown safety issues lead to high failure rates in the drug candidate selection process due to unsystematic evaluation [70]. The absorption, distribution, metabolism, and excretion (ADME) of compounds in the body can be predicted using pharmacokinetic and pharmacodynamic modeling of the liver, kidney, blood vessel and heart, and other relevant tissues according to the target organ of the drug. OoCs can offer advantages over 2D cell culture platforms in the exploration of predictive models as discussed in the introduction section.

## 10. Liver Models

The liver is one of the first organs to experience drug-induced toxicity. For this reason, compound clearance on the liver must be assessed as early as in the product development phase. However, the recent ban on animal use in toxicity tests that were introduced in Europe resulted in the development of bioartificial liver systems [71]. Species-specific liver-chips using rat, dog, and human-derived hepatocytes interfaced with liver sinusoidal endothelial cells can be modeled with OoC [72]. Hepatic toxicity models mimicking hepatocellular injury, steatosis, cholestasis, fibrosis, and species-specific toxicity are assessed when treated with the drug-candidate compounds. Such OoC devices predict liver toxicity and address the human relevance of liver toxicity found in animal studies [73]. For example, chronic hepatotoxicity testing can be performed using a perfusion-incubator-liver-chip fabricated via soft lithography [74]. This system can assess repeated dosing chronic hepatotoxicity with a perfusion incubator that uses CO_2_ gas pressure to drive the perfusion medium. Liver-on-chip devices benefit from the use of 3D printing. Hollow microchannels fabricated using 3D printing and encapsulated in gelatin methacryloyl hydrogels were used as a substrate to co-culture HepaRG and HUVECs to build a model integrating the blood vessel and the hepatocyte layer [75]. This integrated model provides more in-vivo viability and permeability to perform reliable drug toxicity testing. High-throughput hepatotoxicity screening is also made possible in OoC platforms. A liver-on-a-chip incorporating the 96-well plate specification reconstituted a comprehensive liver microenvironment by co-culturing iPSC-derived hepatocytes, endothelial cells, and monoblasts together [76].

## 11. Kidney Models

Nephrotoxicity is considered a major reason for drug attrition in the pre-clinical phase. Approximately 20% of the drugs fail to pass nephrotoxicity tests and therefore cannot advance through clinical trials. To date, cell culture and animal models have been the workhorse of nephrotoxicity tests [77,78]. Inclusion of biophysical stimuli (in this case shear stress induced by the presence of a flow rate) in liver models was shown to enhance drug efflux and albumin uptake despite the application of flow in a unidirectional or bidirectional manner. The epithelium in kidney proximal tubules is continuously exposed to shear stress, which was successfully mimicked by the use of OoC systems [79]. A flow-inducible OoC device mimicking the dynamic culture of kidney organoids exhibited in-vitro glomerular development by inducing morphological maturation with vascularization [80]. It is also possible to model multiple organs with OoC technology. For example, an integrated liver-kidney-chip allowed the evaluation of drug-induced nephrotoxicity following liver metabolism in-vitro, where co-culture of hepatic and renal cells in a compartmentalized multi-layer was exposed to ifosfamide and verapamil [81]. The metabolites produced by liver metabolism were detected using mass spectrometry and were found to cause pronounced nephrotoxic effects on cell viability, lactate dehydrogenase leakage, and permeability of kidney cells (Figure 7) [82,83].

## 12. Brain Models

The blood-brain barrier comprises endothelial cells, separating the blood from brain interstitial fluids. The endothelial cells also form a physical barrier by tight junction proteins that limit the permeation of ions and hydrophilic agents via paracellular pathways [84]. Brain-on-a-chip models mainly focus on creating (i) 2D cell configuration through structural constraints, (ii) porous membrane interface, and (iii) hydrogel-embedded 3D cell constructs separately to study distinct key tissue functions. 2D cell configurations benefit from the ability to fabricate microchannels with different heights, where neuron growth can be tracked. Microcompartments can isolate axons two-dimensionally by separating soma and axon in a neuron [33,85]. In this way, specific drug testing on axon and soma regions can be performed as well as anatomical studies focusing on the reconstruction of unidirectional axonal growth and myelination. In brain-on-chip devices, two or more cells can be cultured to investigate cell-cell communications via cytokine-mediated stimulants. The measurement techniques include off-chip cytokine level determination, protein-level validation, and imaging intracellular Ca2+ levels [37,86]. OoCs have also been built for constructing 3D neurovascular units using collagen and Matrigel, where the passage of small molecule drugs through the BBB was measured on the neurovascular unit [87]. Compound transport efficacy, molecular pathways, and toxicity of neuroactive drugs have been studied on such platforms. Metabolic fluxes and conversions through this neurovascular unit can analyze the role and response of the specific cell types found in the brain. Brain-on-chip devices can also be operated in a high-throughput manner by connecting the devices to high-content screening equipment [88,89].

## 13. Respiratory Models

Lung-on-a-chip was a monumental achievement in the development of OoC. In 2010, an OoC device that reconstructs the functional alveolar-capillary interface of the human lung was demonstrated [1]. This interface provides a comprehensive response at the organ level to bacteria and inflammatory cytokines thanks to the layered human alveolar epithelial and lung microvascular endothelial cells on the membrane. From a toxicological study point of view, the lung-mimicking model revealed that the cyclic mechanical strain featured the lung’s toxic and inflammatory responses to silica nanoparticles. Subsequently, the research group developed a lung-on-a-chip targeting pulmonary edema as a disease model. This chip mimicked the alveolar-capillary interface of a human lung, reproducing drug toxicity–induced pulmonary edema observed in human cancer patients treated with interleukin-2 (IL-2) at similar doses in the same time frame [90]. Mechanical stimulation associated with physiological breathing motions plays a crucial role in the development of increased vascular leakage that leads to pulmonary edema. After formulating the molecular pathways, OoC devices have been used for the identification of potential new therapeutics, including angiopoietin-1 (Ang-1) and a new transient receptor potential vanilloid 4 (TRPV4) ion channel inhibitor (GSK2193874), which might prevent this life-threatening toxicity of IL-2.

## 14. Cardiovascular Models

The tissues in the cardiovascular system experience various types and levels of biophysical stimuli. For example, heart tissue experiences shear stress, tensile strain, and stretching when cardiac muscles pump blood into the vessels. Similarly in vessels, endothelial cells surrounding the inner wall of the vasculature are exposed to similar types of biophysical stimuli as well as hydrostatic pressure due to the pulsatile flow created by the heart. Among these stimuli, shear stress was found to be the most significant one, influencing cell morphology and proliferation characteristics (Figure 8). OoC models of the cardiovascular system have been studied with a focus on blood vessels and heart-on-chip platforms because deformation of the vascular barrier has a central role in many cardiovascular diseases. Once the vascular barrier architecture is created, many research questions could be studied on these platforms. For example, the adhesion of neutrophils inside lung microvessels was studied to mimic chronic obstructive pulmonary disease conditions. Shear stress at different pressure levels along with cyclic membrane stretching can also be applied on the microchannels to initiate rapid hematopoietic cell formations [91].

## 15. Intestine Models

The intestine is one of the organs with complex mechanics. Peristaltic motion is a highly synchronized contraction movement, which results in irregular tensile and compressive strains as well as shear stress. Such biophysical stimuli changes in different parts of the intestine because the viscosity of the medium keeps changing along the intestine as the digestion process continues. Intestine models have been a popular topic in OoC models because the intestine is the primary place to absorb drugs. A range of disease models including inflammatory bowel disease and colitis is interesting for drug toxicity testing [92]. Peristaltic motions in OoCs are generated in the form of cyclic stretching of PDMS, shear stress is generated via fluid flow in microchannels while co-culture of intestinal cells is performed using e.g., hydrogels or microchip architecture (Figure 8) [93]. In OoCs, epithelial barrier integrity, viability, mucus bilayer formation, and bacterial infection topics are studied as key parameters defining how good the mimicking conditions are [94,95,96,97]. In this context, OoCs do not focus on mimicking the entire intestine, but rather parts of it such as the small intestine, duodenum, and colon.

## 16. Musculoskeletal Models

The type of biophysical stimuli experienced by the musculoskeletal system changes depending on location. While connective tissues are exposed to extreme stretching, the response of the cells varies according to the stimuli [98]. Articular cartilage is an interesting tissue for OoCs as just a normal physical activity can create compression, shear stress, tensile stress, and osmolarity effects. Inappropriate application of forces results in osteoarthritis and tendinopathy that can also be modeled in OoC devices [99]. On the other hand, the musculoskeletal system is one of the most challenging and overlooked applications of OoC given the fact that the complex formation of tissues and biophysical stimuli. In a bone-marrow-on chip application, cell-seeded hydroxyapatite-coated zirconium oxide scaffolds maintained long-term culturing of multipotent hematopoietic stem and progenitor cells under fluid flow (Figure 8) [100].

**Figure 8 micromachines-13-01200-f008:**
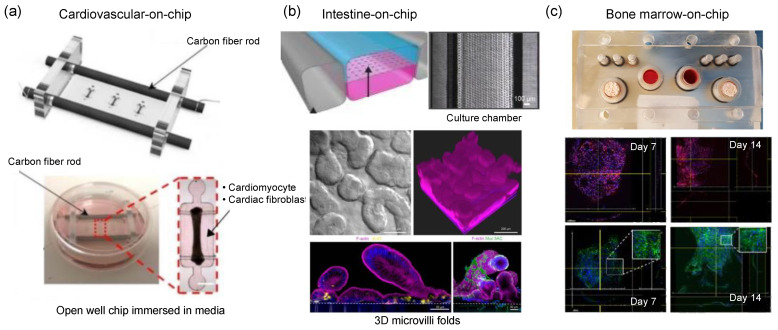
Cardiovascular, intestine and musculoskeletal OoC models. (**a**) Open well OoC device with carbon fiber rods and platinum wires. The voltage applied across the open well tissue culture-induced cardiac contractions. Figure reproduced from Mastikhina et al., 2020 [91]. (**b**) The primary human intestine chip contained human intestinal epithelium and intestinal microvascular endothelium co-cultured while the peristaltic motion was mimicked via an elastic membrane. Figure reproduced from Kasendra et al., 2018 [93]. (**c**) The bone-marrow on-chip model is cultured for up to four weeks under fluid flow. Figure reproduced from Sieber et al., 2018 [100].

## 17. Multi-Organ-on-Chip

Organ-to-organ interactions play a key role in identifying the toxicity of drugs. For example, a drug with high toxicity levels may be detoxified in the liver and converted to non-toxic compounds [101]. Conversely, a harmless drug may be metabolized in the liver into metabolites that are toxic to other tissues. OoC devices facilitate functional combinations such as modeling multiple organ ADMEs. OoCs are linked to each other using the elements required to scale the target organs, for example, dynamic flow fluid, cell composition, culture time point, and cellular compartment [102,103].

Recently, a tissue chip system connected by circulating vascular flow has been developed. In this system, the human heart, liver, bone, and skin tissue niches are cultured in their respective chambers and the endothelial barrier connecting them allows for interdependent functions. The interconnected tissues recapitulated the pharmacokinetic and pharmacodynamic profiles of doxorubicin and identified miRNA biomarkers [104]. A fully automated cell culture system was developed, interconnecting devices capable of continuous perfusion, medium exchange, fluid connection, sample collection, and in situ microscopy [105]. In another multi-organ microfluidic ‘physiome-on-a-chip’ platform as shown in Figure 9, [106]. Ten OoCs were linked to each other for 4 weeks to study pharmacokinetic analysis of diclofenac metabolism. Another multi-OoC was used for modeling anti-leukemia drug analysis on cancer-derived human bone marrow for investigating the toxicity effect of the liver using a multi-organ configuration. Similarly, multi-OoCs were used for studying multidrug-resistant hepatocytes and induced pluripotent stem cells–derived cardiomyocytes (Figure 9). Such pharmaceutical testing systems can establish a therapeutic window for comprehensive compound evaluation in human tissues. The accuracy of these systems is determined by evaluating repeated dose effects and off-target effects.

## 18. Commercialization

Since OoC attracted the biomedical research community over a decade, the increasing interest has triggered the commercialization of the technology. Over 20 start-up companies including spin-offs from academic OC research groups actively supply products or provide services to OoC users [7,107,108]. The majority of the commercial systems incorporate microfluidics for the handling of cell microenvironment, therefore also introducing biophysical stimuli in the form of shear stress and interstitial flow. The minority of the commercial systems can produce mechanical strain such as breathing or peristalsis motion. The end-users of this commercialized platform range from individual researchers to pharmaceutical companies. Most of the commercialized products currently focus on drug toxicity testing and safety validation. Scaling-up in fabrication, validation of the reproducibility of OoC, and user-friendliness in the design of OoC devices are the major challenges to overcome before adopting the OoC technology in drug discovery phases.

## 19. Limitations

The technical level of OoC has reached the state of art, but the real-world application of the platform remains a challenge. Validation of reproducibility and accuracy of the OoC models, compared to in-vivo models or clinical trials, is required for full integration of the OoC technology into the drug discovery phases. Moreover, improved high-throughput operation and feasibility of mass production are key to reaching out to various end-users from pharma-industry associates to fundamental research scientists [109].

Innovative advancements have been achieved in the field of OoC in the past decade, while most of the models are developed as proof-of-concept. Bench-top-produced devices are a major bottleneck for commercial use in drug discovery research. PDMS-based OoC fabrication remains the most popular and traditional method to date, requiring dedicated fabrication equipment such as photo- and soft-lithography stations, clean bench stations, and an oxygen plasma treatment machine. During the device construction, liquid and hydrogel patterning inside the microfluidic channels require trained personnel due to the complexity in the design of the current OoC models [110]. Complications associated with microfabrication lead to poor interfacing, laborious production, and the burden of expenses for dedicated equipment.

Standardization is another problem being tackled in current OoC technology [111]. Reported OoC models vary extremely in the design and generation of physiologically relevant conditions. Although the OoC models are tested for reproducibility, it is difficult to control user-to-user reproducibility. Microfabrication and tissue culturing methods differ widely between the users, for example, the dimension of the microchannels, the initial number of cells in (co-)culture, and type of 3D matrix, etc. The standardization cannot be easily established in OoC devices as it was achieved for 2D cultures, because OoC devices have much more components and functionalities compared to 2D culture plates.

Analytical outcomes also lack full validation in OoC systems. Due to the performance variation of the monitoring tools, the range of the effective dosage of the drug applied in an OoC device may differ from in-vivo [72]. Scalability (i.e., analyzing the results from micron-scale environments for macro-scale systems) is also seen as a problem in OoC devices. The standardized references should be set up for analogical interpretation of drug test results from OoC. Compound concentrations are difficult to measure in PDMS-based platforms due to the adsorption of the small molecules into the PDMS material [12,112,113].

## 20. Future Considerations

The biggest expectation from OoC technology relies on its application in drug discovery phases. OoC devices successfully complement traditional models, while the simplicity and physiological relevance with organ-level complexity could be balanced depending on the purpose of a study. For instance, high-throughput, drug-toxicity-testing OoC models are expected to adopt a relatively simple structure similar to medium-throughput models designed to study advanced pathological functions in human disease [114].

OoC and organoids essentially have the same goal of recapitulating the functional and morphological properties of human organs in vitro. However, OoC technology relies on the development of artificial models to build structurally well-established systems in which cells and microenvironments can be precisely controlled. In contrast, organoids develop from autologous stem cells to conform a developmental program and reproduce key structural and functional properties of their in vivo architecture. Recently, as the concept of organoid-on-a-chip has emerged, the development of a platform that can help the maturation of organoids and further expand to large-scale experiments such as drug screening is becoming an important research field. It is now necessary to develop a platform that is more reproducible and more controllable through OoC technology rather than the classical organoid culture approach, extending it into the pipeline of drug discovery [115].

The critical point in adopting OoC in translational medicine is considering the phase of preclinical research progress. The early stage of the preclinical research, such as discovering drug candidates, would require an elaborate recapitulation of human physiology, which will enable the validation of the target drug mechanism. Mid-stage preclinical progress for narrowing down the specific target or dosage of the drug would require more strength in high-throughput property of the OoC to screen the efficacy efficiently. Multi-organ-on-chip could play on the last step of the preclinical studies to verify the toxicity and to provide better insight into the pharmacokinetics of the drug candidate [116].

Moreover, OoC models can facilitate personalized-medicine applications using cells obtained from real patients. In other words, OoC models can serve as a tool to predict the patient-specific response of a drug by interacting with patient-derived primary cells or stem cells in the cell microenvironment [6,117,118,119]. The personalized medicine approach is currently a popular application of OoC devices, and the validation and standardization studies are conducted accordingly.

Beyond the application of OoC in drug discovery, the incorporation of OoC models with gene editing strategies such as the CRISPR-Cas9 molecular system could help in the screening of cell phenotypes [2,72,111]. Moreover, omics-based analytics including proteomics and single-cell gene sequencing could be also implemented in OoC devices as monitoring tools [120,121,122].

## 21. Conclusions

OoC models serve as innovative tools to investigate the key functions of human tissues. These models are good alternatives to traditional preclinical models such as in-vivo animal models and simple 2D in-vitro models in the drug discovery process. Currently, the OoC technology is in active expansion to biomedical research, especially personalized medicine, while commercialization strategies are being applied by fulfilling the requirements of users [2,5,6,107,110,122].

Successful application of OoC devices requires a clear understanding of each component. Researchers interested in working with OoC models must be educated on several topics including materials, microfabrication methods, and structural design of the chip depending on the purpose of the study. For instance, plastic materials must be preferred when studying drug toxicity screening on simple tissue culture models [12,111,112,113]. If PDMS material is chosen for such a study, small molecule adsorption will limit the reliability of the model. Instead, PDMS will be handy when modeling complex, multi-layer tissue structures composed of a multi-cellular microenvironment. The selection of cell sources is another key point in developing OoC with highly physiological relevance. As the application of OoC expands to the biomedical and pharmaceutical field, usage of patient-derived cells and iPSCs is encouraged when recapitulation of patient-specific phenotype is needed [118,121,123,124,125]. The incorporation of biochemical and biophysical stimuli is also a critical factor to improve the functionality of the engineered tissues in OoCs [76]. During the construction of OoC devices, multiple monitoring techniques are integrated to collect real-time data which relates to the specific function or phenotype of the modeled tissue.

The birth of OoC technology stems from two major purposes. First, in-vitro models with patient-derived cells recapitulating tissue microenvironment or physiological function are introduced for studying the biology behind disease and pathology. Conventional animal models bear the risk of interspecies difference-based alterations in experimental results as well as the low-throughput nature of the models. Second, OoC can be used in the drug development process, both in drug toxicity and efficacy tests. The level of complexity in OoC models can be adjusted according to the research question and the phase of translational research. Particularly, recent trends in OoC research take attempts for commercializing high-throughput OoC models, which is appealing to pharmaceutical companies [5,7,126].

From developers to users, the OoC community grows rapidly. The interdisciplinarity of the community is the key factor for the successful translation of OoC to bench-to-bedside translation research.

## Figures and Tables

**Figure 1 micromachines-13-01200-f001:**
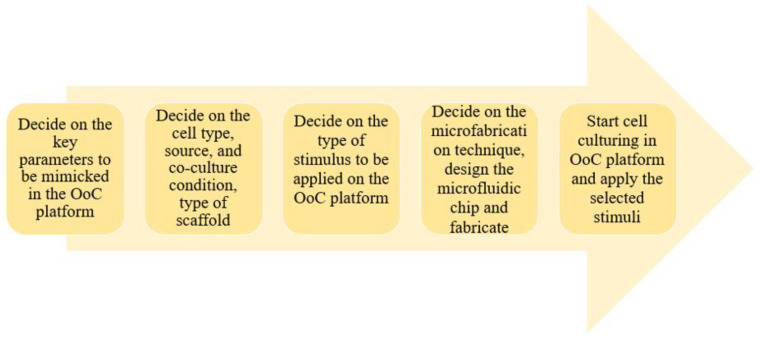
Typical prototyping steps followed to fabricate a microfluidic OoC platform.

**Figure 4 micromachines-13-01200-f004:**
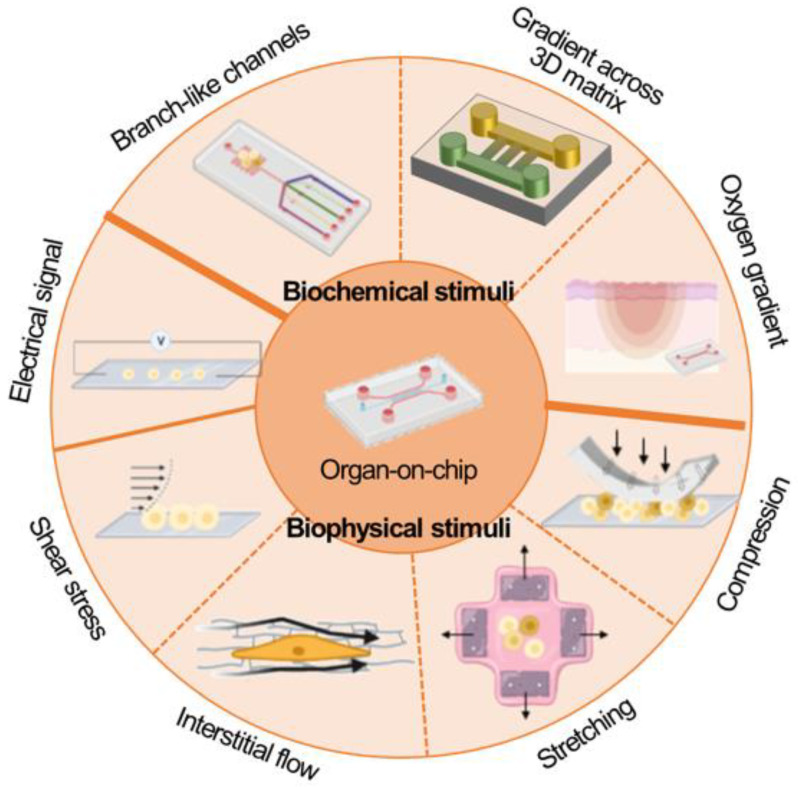
Biochemical and biophysical stimuli applied to organ-on-chip models.

**Figure 5 micromachines-13-01200-f005:**
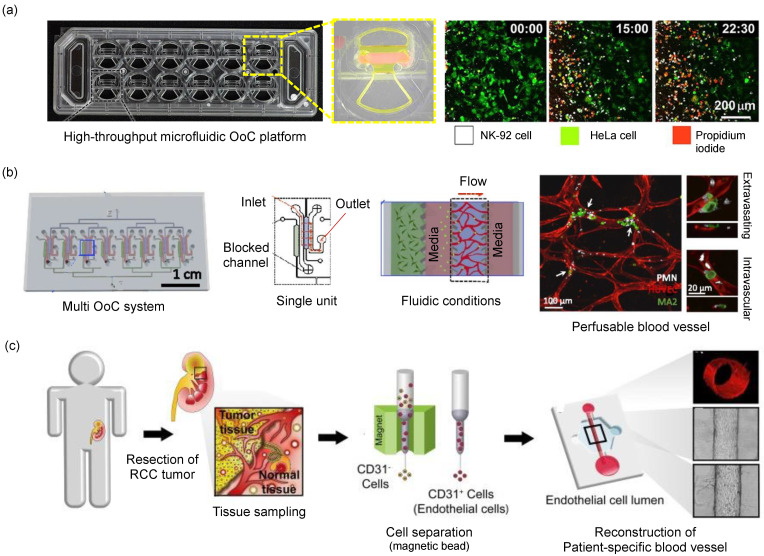
Modeling tumor microenvironment using microfluidic devices. (**a**) Large-scale producible microfluidic chip to utilize 3D cytotoxicity assay. Figure reproduced from Park et al., 2019 [21]. (**b**) Multiplexed microvascular network to quantify the dynamics of arrest and extravasation of tumor cells and inflammation-stimulated neutrophils in the microfluidics chip. Figure reproduced from Chen et al., 2018 [61]. (**c**) Isolation endothelial cells from cancer patient samples and reconstruction of patient-specific tumor vasculature models in the microfluidic device. Figure reproduced from Jimenez-Torres et al., 2019 [62].

**Figure 6 micromachines-13-01200-f006:**
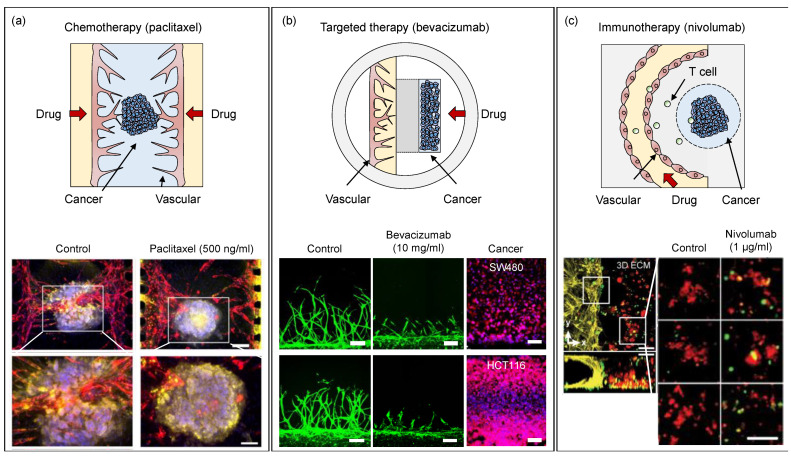
Examples of cancer-vascular models with anticancer drug screening. (**a**) On-chip tumor spheroid and blood vessel co-cultured microenvironment with biochemical and biophysical factors. Figure reproduced from Nashimoto et al., 2020 [67]. (**b**) A standardized microfluidic platform for high-throughput anti-angiogenic drug screening. Figure reproduced from Kim et al., 2021 [17]. (**c**) OoC device mimicking glioblastoma tumor niche including immune cells, brain microvessels, tumor-associated macrophages, and glioblastoma. Figure reproduced from Cui et al., 2020 [69].

**Figure 7 micromachines-13-01200-f007:**
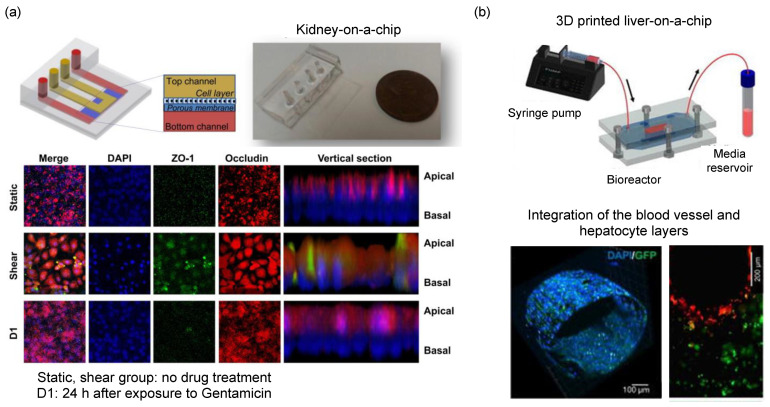
Liver and kidney-on-a-chip for hepatotoxicity and nephrotoxicity testing. (**a**) A study showing a pharmacokinetic profile of reducing nephrotoxicity of gentamicin in perfused kidney on-chip. Figure reproduced from Kim et al., 2016 [79]. (**b**) 3D printed liver-on-chip device for evaluating in-situ hepatocyte cell differentiation. Figure reproduced from Massa et al., 2017 [75].

**Figure 9 micromachines-13-01200-f009:**
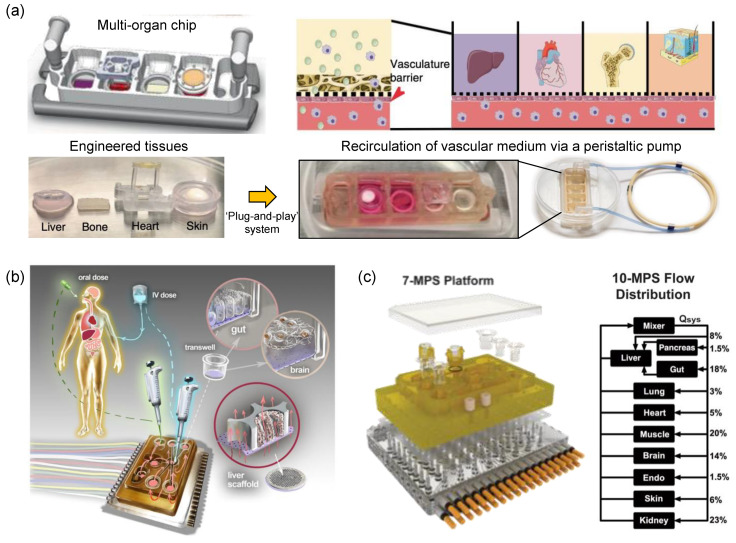
Modeling multi-organ systems on a chip. (a) The integrated multi-organ chip connects the liver, bone, heart, and skin along the vascular flow with the concept of a plug-and-play system to maintain tissue-specific niche. Figure reproduced from Ronaldson-Bouchard, K. et al., 2022 [104]. (**b**) Schematic diagram of a multi-organ on-chip system [106]. (**c**) Design and construction of multi-organ partitions within the microphysiological system platform. Multi-organ flow distribution design through the arrangement of major components such as channels, pumps, and reservoirs. Figure reproduced from Edington et al., 2018 [106].

**Table 1 micromachines-13-01200-t001:** Comparison between advantages and disadvantages of the available translational research model.

Research Approach	Macro-Scale Techniques	Micro-Scale Techniques	Animal Models
Platform	Well plates	Microfluidic chips	Experiment animals
Types	2D cell culture Spheroid assay	Organ-on-a-chip	mouse, rabbit, monkey, murine, etc.
Advantages	A standardized platform, one size fits all principle Well established process flow Co-culture of different cell types Cost-effective	Custom-designed per case Precise control of cell microenvironment Co-culture of different cell types Fluid flow application Dynamic cell-culture environment Possibility to integrate with stimuli sources and measurement tools such as sensors Reduced use of the reagents	Physiologically relevant results Availability of in vivo conditions Availability to observe the response of the organism as a complex entity
Disadvantages	Static (no flow) conditions Physiologically less relevant model Uniform distribution of reagents (i.e., no possibility to form gradients) One culture condition can be tested at a time Well plates are made of only one material, which is polystyrene	Commercially not available Non-standardized process flow Microfabrication requirement for prototyping Not as budget friendly as the conventional cell culture platforms	The studies are subject to ethical concerns Expensive and hard to maintain Not possible to apply real-time monitoring in the absence of dedicated equipment Complicated platform for mechanistic studies

**Table 2 micromachines-13-01200-t002:** Comparison of existing preclinical research models and OoC devices.

	Animal Model	2D Cell Culture	OoC Device
Human tissue	No	Yes	Yes
Personalized medicine	No	Yes	Yes
Complexity	Yes	No	Limited
Control over microenvironment	No	Yes	Yes
Tissue-level function	Yes	Limited	Yes
Organ-level function	Yes	Limited	Limited
Real-time readouts	No	Limited	Yes
High-throughput, in parallel testing	No	Yes	Yes
Pharmaco-dynamics and-kinetics	Yes	No	Yes

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
