# Peer review of "Engineering Organ-on-a-Chip to Accelerate Translational Research"

_micromachines, 2022, doi:10.3390/mi13081200_

Round 1

Reviewer 1 Report

This is a relatively comprehensive review that introduces the emerging field of organ-on-a-chip and some design considerations in this field. It can be helpful for general researchers interested in working in this field. I have the following suggestions to improve the quality of the manuscript:

1)    As the title of the article suggested, one would expect to see more statements on how organ-on-a-chip can accelerate translational research. It will be more beneficial if the authors introduce the various stages of the translational research, its challenges and how microfluidic systems can contribute to this field.

2)     There are many interesting review articles in this field. The authors need to refer to the notable ones in the introduction and clearly state how the current work would differ from them.

3)     Another topic that can be further expanded in this review is oxygen control using organ-on-a-chip devices. There are interesting reviews and research articles that illustrate how microfluidic systems can control the oxygen concentration within the desired level of physiological or pathophysiological (for diseases modelling) conditions. In particular, when using thermoplastic materials instead of PDMS. Please refer to the following studies for more details:  DOI:10.1039/C4LC00853G; 10.1007/s10544-022-00615-1

4)     The examples provided in Figure 6 for cancer-vascular models with anti-cancer drug screening are not clearly showing this concept. As a minor comment, the quality of figure 6A is unacceptably low and should be replaced.

5)     There are many engaging microfluidic platforms for anti-cancer drug screening that, at the same time, can prevent the issue of air bubbles in PDMS-based microfluidic devices. Please refer to the following review papers for more details: 10.1016/j.trac.2020.116118 and 10.3390/app11209418

Reviewer 2 Report

Recommendation: Minor revisions needed as noted.

This manuscript seems to have a good theme and a well-made structure as a review paper. Taking into account the quality of work and scope of the journal, I would recommend the minor revision according to the following comments. Some suggestions were listed as follows for the authors to improving the work.

 Comments:

 # 1. I suggest the authors to describe the differences between organ-on-a-chip and organoids chip for the clear understanding of the readers.

 # 2. Although the research topic raised by author is interesting and crucial in bioengineering fields, I am afraid that the current version of the manuscript has many typos and sentences are incomplete. This makes it difficult for the readers to fully appreciate the work presented.

- Ca2+ levels, 10. Drug Development, Fabrication of spheroid organoid cultures, etc.

 # 3. Some font sizes in figures (especially figs. 5 - 8) are way too small to read. Please manually handle the high-resolution figures and increase the text visibility.
